# Individual and population level costs and health-related quality of life outcomes of third-generation cephalosporin resistant bloodstream infection in Blantyre, Malawi

**Rebecca Lester** [1,2,3] *, **James Mango** [2], **Jane Mallewa** [2,4], **Christopher P. Jewell** [5], **David A. Lalloo** [1], **Nicholas A. Feasey** [1,2], **Hendramoorthy Maheswaran** [6]

1 Department of Clinical Sciences, Liverpool School of Tropical Medicine, Liverpool, United Kingdom,
2 Malawi Liverpool Wellcome Research Programme, Kamuzu University of Health Sciences, Blantyre,
Malawi, 3 Division of Infection and Immunity, University College London, London, United Kingdom,
4 Department of Medicine, Kamuzu University of Health Sciences, Blantyre, Malawi, 5 Centre for Health
Informatics, Computing and Statistics, Lancaster University, Lancaster, United Kingdom, 6 Institute of Global
Health Innovation, Imperial College London, London, United Kingdom

* r.lester@ucl.ac.uk

**Data Availability Statement:** The original data used for this project are available at Zenodo, DOI

## Abstract

Data which accurately enumerate the economic costs of antimicrobial resistance (AMR) in low- and middle- income countries are essential. This study aimed to quantify the impact of third-generation cephalosporin resistant (3GC-R) bloodstream infection (BSI) on economic and health related quality of life outcomes for adult patients in Blantyre, Malawi. Participants were recruited from a prospective, longitudinal cohort study of hospitalised patients with bloodstream infection caused by Enterobacterales at Queen Elizabeth Central Hospital (QECH). Primary costing studies were used to estimate the direct medical costs associated with the inpatient stay. Recruited participants were asked about direct non-medical and indirect costs associated with their admission and their health-related quality of life was measured using the EuroQol EQ-5D questionnaire. Multiple imputation was undertaken to account for missing data. Costs were adjusted to 2019 US Dollars. Cost and microbiology surveillance data from QECH, Blantyre was used to model the annual cost of, and quality-adjusted life years lost to, 3GC-R and 3GC-Susceptible BSI from 1998 to 2030 in Malawi. The mean health provider cost per participant with 3GC-R BSI was US\$110.27 (95%CR; 22.60–197.95), higher than for those with 3GC-S infection. Patients with resistant BSI incurred an additional indirect cost of US\$155.48 (95%CR; -67.80, 378.78) and an additional direct non-medical cost of US\$20.98 (95%CR; -36.47, 78.42). Health related quality of life outcomes were poor for all participants, but participants with resistant infections had an EQ-5D utility score that was 0.167 (95% CR: -0.035, 0.300) lower than those with sensitive infections. Population level burden estimates suggest that in 2016, 3GC-R accounted for 84% of annual societal costs from admission with bloodstream infection and 82% of QALYs lost. 3GC-R bloodstream infection was associated with higher health provider and patient level costs than 3GC-S infection, as well as poorer HRQoL outcomes. We demonstrate a substantial current and future economic burden to society as a result of 3GC-R *E. coli* and

[10.5281/zenodo.7588818](https://doi.org/10.5281/zenodo.7588818) Data are available under the terms of the Creative Commons Attribution 4.0 International license (CC-BY 4.0).

**Funding:** This work was supported by the Wellcome Trust (Clinical PhD Fellowship, University of Liverpool block award grant number [203919/Z/16/Z] (RL)). The funders had no role in study design, data collection and analysis, decision to publish, or preparation of the manuscript.

**Competing interests:** The authors have declared that no competing interests exist.

*Klebsiella* spp. BSI, data urgently needed by policy makers to provide impetus for implementing strategies to reduce AMR.

## Introduction

The funding and resources required to mitigate the impact of antimicrobial resistance (AMR) are likely to be huge [1, 2], and need to be backed by accurate enumeration of costs to understand the efficiency of these investments and develop appropriate policy responses for treatment and prevention. These economic considerations are of particular importance in low income countries [3]. The economic costs of AMR are predicted to be highest in sub-Saharan Africa (sSA), but data from Low and Middle Income Countries (LMICs) to support these predictions are lacking [2].

In Malawi, public health services are provided freely to residents and medical care at hospitals is free at the point of delivery. Nonetheless, medical admissions incur costs to the health provider and to the patient and these costs need to be understood for budgetary planning and informing decisions around interventions to reduce AMR. In Malawian hospitals, third-generation cephalosporins (3CG) are the antibiotic of choice for the empirical management of suspected sepsis or any severe bacterial infection, however there has been rapid proliferation of 3CG resistance (3GC-R), and alternatives are lacking. The change to these agents circa 2004 was based on ease of administration (once daily) and affordability (donor funded) as well as antimicrobial susceptibilities at the time. This study aimed to quantify the impact of 3GC-R bloodstream infection (BSI) on economic and health related quality of life outcomes for adult patients admitted to the adult medical wards at Queen Elizabeth Central Hospital (QECH) in Blantyre.

Specific objectives included; estimating healthcare provider costs of providing inpatient medical care to adult patients with 3GC-R BSI at QECH (direct medical costs). Secondly, to estimate the costs to individuals who are admitted to QECH with 3GC-R BSI (direct non-medical and indirect costs). Thirdly, to estimate the impact of 3GC-R BSI on the health-related quality of life (HRQoL) of adult patients admitted to QECH. Lastly, to use the cost and HRQoL estimates to explore the economic burden posed by 3GC-R BSI in Malawi.

## Methods

### Study design, setting and data collection

This was a sub-study of a prospective longitudinal cohort of patients at QECH who had bloodstream Enterobacterales, protocol detailed at [4]. Malawi is a low-oncome country with low healthcare expenditure (~ 3% GDP) and an estimated population of 17.5 million people Blantyre is the second city and commercial capital, with a population of 800,264. QECH is a 1300 bedded teaching public hospital which provides free inpatient secondary care to Blantyre and tertiary care to surrounding districts. It receives approximately 10,000 adult admissions per year. The Human Immunodeficiency Virus (HIV) prevalence amongst medical inpatients was at least 50.0% between 2012 and 2019 and 79.7% of those known to be HIV-positive on admission were on antiretroviral therapy (ART) [5]. The hospital has a large emergency department where all new patients are triaged and assessed by medical doctors or clinical officers. Clinicians make a preliminary medical diagnosis, and those in need of admission are transferred to one of three medical wards (Male Medical; Female Medical; Tuberculosis (TB) Ward).

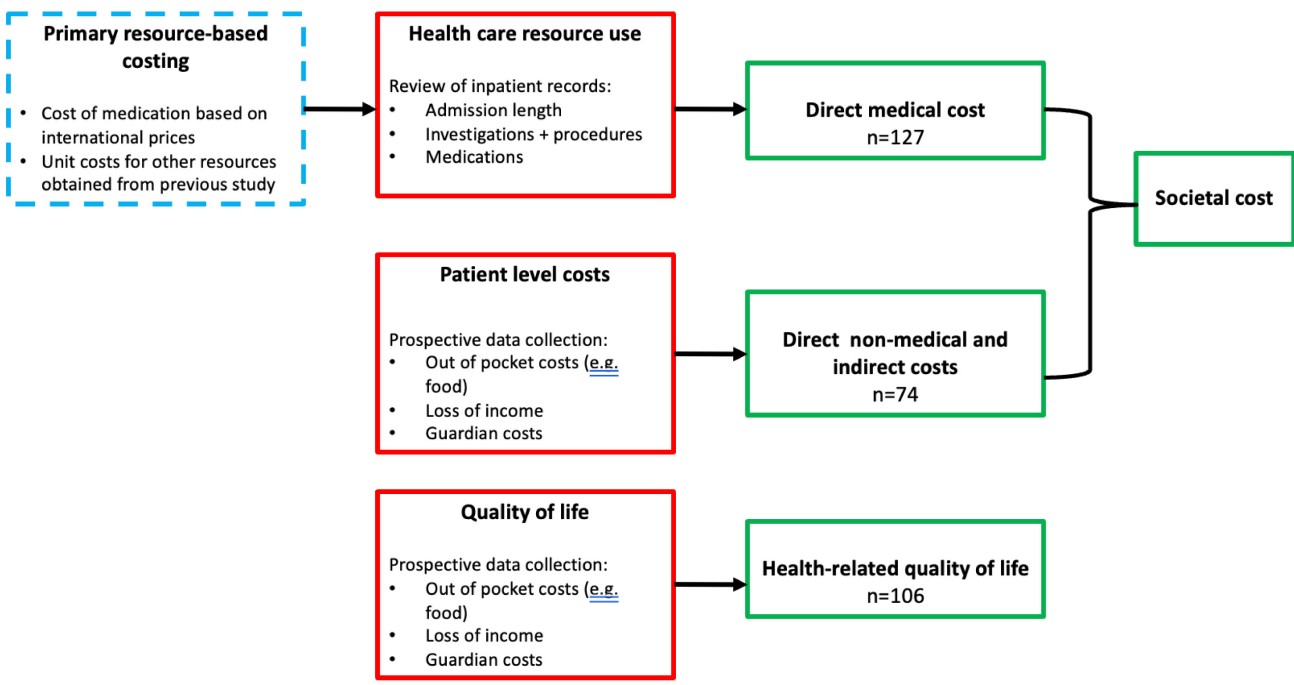

**Fig 1. Overview of data collection and health economic analysis.** Blue dashed box indicates data collected from previous costing study at QECH [7]. Red boxes describe raw data collection and green boxes describe health economic outputs. Numbers in green boxes indicate number of participants included in each estimate. The sample sized denoted in the green boxes refer to the number of actual datasets available. Model predictions were performed on a total sample of 154 participants and multiple imputations done to impute missing values.

An overview of the data collection methods is shown in Fig 1 and the CRFs used are included as extended data in the published protocol [4]. To summarise, data were collected on medical diagnosis and resource use, and primary resource-based costing studies to estimate health provider costs. Costs incurred by patients and their families as a result of hospitalisation, were also investigated, and the health-related quality of life (HRQoL) of participants was evaluated. Costing comparisons were made between participants who had 3GC-R and 3GC-S infection.

The detailed laboratory methods are described elsewhere, but briefly, blood cultures were processed at the ISO accredited, quality assured diagnostic laboratory at MLW (UK National External Quality Assessment Service. Antimicrobial susceptibility (AST) was established using the disc diffusion method and EUCAST breakpoints (EUCAST version 7.0) [6]. 3GC-R was defined as resistance to one or both of cefpodoxime or ceftriaxone and was confirmed using the species dependent combination disc method.

## Direct medical costs

The study clinician reviewed the medical notes upon discharge or death to capture healthcare resources used during each participants medical admission. A structured questionnaire recorded type and quantity of each resource used, including medications, investigations and procedures, as well as the duration of hospital admission. For medications, including intravenous fluids, the study clinician recorded the route of administration, dosage and number of doses given. The international market price was used for the cost of medications and intravenous fluids [8]. The unit costs for remaining healthcare resource item were obtained from a

previous resource-based hospital costing study undertaken at QECH in 2014 [7] and adjusted to 2019 US Dollars using data reported by the World Bank [9].

### Direct non-medical and indirect costs

A previously developed questionnaire [4, 10] was used to collect data to estimate the direct non-medical and indirect costs incurred by each participant and their guardian during the hospital admission [7]. This questionnaire was administered to patients and their guardians as close to discharge as possible and ideally on the day of discharge. Data collected included cost of transportation, food, drinks, toiletries, clothing and other items bought during the hospital admission. The total of all these costs equated to the direct non-medical cost.

For indirect costs, the time off work taken by participants and their guardian (in days), was multiplied by their self-reported daily income [11]. These direct non-medical costs and indirect costs were estimated in Malawian Kwacha (MWK); and converted to 2019 US Dollars using the prevailing exchange rate at time the analysis was undertaken (1US$ = 750 MWK).

**Societal costs.** Societal costs refer to the sim of direct medical, direct non-medical and indirect costs (Fig 1). Costs were estimated in Malawian Kwacha (MWK); and converted to 2019 US Dollars using the prevailing exchange rate at time the analysis was undertaken (1US$ = 750 MWK).

### Health-related quality of life (HRQoL)

The Chichewa version of the EuroQoL EQ-5D-3L was used to capture the HRQoL of participants recruited into this study [12]. EQ-5D has a descriptive component and a visual analogue scale (VAS) [4]. The descriptive component assesses HRQoL across five domains: anxiety, pain, self-care, usual activities and mobility. Participants rate themselves on a 3-point ordinal scale (no problems, moderate or extreme problems). Responses to the measure would generate 243 unique health states, each one then converted to an EQ-5D utility score using a tariff set. Tariff sets have been derived for several countries through national surveys of the general population. No tariff set currently exists for Malawi and therefore the Zimbabwean tariff set was used to calculate the EQ-5D utility scores [13]. EQ-5D utility scores range from 1.0 (representing perfect health) to -0.29, negative EQ-5D utility scores equate to health states the general population considers worse than death. As a sensitivity analysis, the UK tariff set was also used to generate EQ-5D utility scores [14]. The visual analogue scale is similar to a thermometer, and ranges from 100 (best imaginable health state) to 0 (worst imaginable health state). Participants record how good or bad their health is on that day by drawing a line on the scale.

### Statistical analysis

Multiple imputation using chained equations was undertaken to impute missing values for cost and health estimates for participants for whom data was missing [15]. We assumed data was missing at random, and our imputation models included age, sex, socio-economic variables, HIV status, organism causing the BSI and 3GC-R status. Predictive mean matching was used to impute missing values for cost and HRQoL estimates as they were nonnormally distributed and to prevent negative estimates being imputed for costs [16].

The total direct medical cost per participant was estimated by summing the cost of hospital ward stay, the cost of all investigations and procedures and the cost of all medications given. The cost of hospital stay was calculated by multiplying the daily cost of admission by length of admission in days. The costs of investigations and procedures was estimated by multiplying the unit cost of each investigation/procedure by the number of times it was performed during

the hospital stay. The cost of all medications given was estimated by multiplying the cost of each individual drug by the number of doses used.

The total direct non-medical and indirect costs per participant was estimated by summing the costs incurred by the participant and their guardian. The total societal cost per participant cost was estimated by adding the total direct medical cost, and the total direct non-medical and indirect cost.

EQ-5D utility and VAS scores were calculated for each participant. For the primary analysis, we present the EQ-5D utility scores generated using the Zimbabwean tariff. For the sensitivity analysis we present the EQ-5D utility scores generated using the UK tariff.

The mean total direct medical costs, direct non-medical and indirect costs and EQ-5D utility scores were calculated and stratified by the organism causing the BSI and by 3GC-R status. The means and standard errors are presented for all costs and the cost differences are presented as means with 95% credible intervals (95% CR).

Multivariable analysis was then undertaken to explore the independent effects of 3GC-R on total direct medical cost, HRQoL outcomes and total societal costs. For each multivariable analysis, two alternate models were constructed. The first was adjusted for organism, age and sex. The second model was additionally adjusted for HIV status as previous work found that HIV infection was associated with higher costs of healthcare at QECH [7].

The cost and EQ-5D utility data were non-normally distributed and therefore non-parametric bootstrapped methods were used to generate robust standard errors and 95% credible intervals. For the multivariable analysis of costs we used generalised linear models (GLM) and ran model diagnostics to determine optimal choices for distributional family and link functions, for the multivariable analysis of EQ-5D utility scores we evaluated commonly used estimators (ordinary least squares; Tobit, fractional logit and censored least absolute deviations) and applied a range of tests to evaluate whether one estimator consistently provided more accurate estimates [17–19].

The direct medical cost, societal cost and EQ-5D utility data was used alongside microbiology surveillance data (1998 to 2016) for the city of Blantyre [20] to model the annual cost of, and quality-adjusted life years lost to, 3GC-R and 3GC-S BSI for 1998 to 2030 in Malawi. The future projections accounted for population growth but assumed incidence of 3GC-R and 3GC-S BSI remained at comparable levels to last year surveillance data was available. More detailed description of methods can be found in S1 Text.

Analysis was carried out in Stata version 13 (Stata Corporation, Texas, USA). Figures were generated using R (version 4·0·2, R Foundation for Statistical Computing, Vienna, Austria).

### Ethics

Ethical approval for the study was granted by the University of Malawi College of Medicine Research Ethics Committee (COMREC), protocol number P.10/17/2299 and by the Liverpool School of Tropical Medicine (LSTM) Research Ethics committee, protocol number 17–063. LSTM acted as study sponsor. Written informed consent was obtained from study participants or their guardian if the patient lacked capacity to consent.

### Results

A total of 154 participants were recruited into this study (Fig 1). Direct medical cost data were available for 127/154 participants, with the remainder not included due to missing patient files at the time of discharge or death. Direct non-medical and indirect cost data were available for 74/154 participants with the remainder missing due to patients dying before questionnaires could be administered. EQ-5D utility scores were available for 106/154 participants and EQ-

**Table 1. Participant characteristics (N = 154).**

| | Characteristic | 3GC-S, n = 69 mean (95% CI) | 3GC-R, n = 85 mean (95% CI) |
|---|---|---|---|
| **Age** | Median (IQR) | 44.6 (33.0, 61.5) | 40.0 (29.0, 54.8) |
| **Sex** | Female | 36 (52.2%) | 78 (50.6%) |
| | Male | 33 (47.8%) | 76 (49.4%) |
| **Education** | No Formal Schooling | 0 (0%) | 3 (3.5%) |
| | Any Primary | 34 (49.3%) | 22 (25.9%) |
| | Any Secondary | 12 (17.4%) | 22 (25.9%) |
| | College or higher | 10 (14.5%) | 11 (12.9%) |
| | Not Known | 13 (18.8%) | 27 (31.8%) |
| **Employment** | Paid employee | 6 (9.4%) | 12 (14.6%) |
| | Paid domestic worker | 1 (1.6%) | 0 (0%) |
| | Self-employed | 18 (28.1%) | 11 (13.4%) |
| | Unemployed | 23 (35.9%) | 29 (35.4%) |
| | Student | 3 (4.7%) | 3 (3.7%) |
| | Other | 13 (20.3%) | 27 (32.9%) |
| **HIV status** | Negative | 24 (34.8%) | 40 (47.1%) |
| | Positive | 44 (62.8%) | 42 (49.4%) |
| | Unknown | 1 (1.5%) | 3 (3.5%) |
| **ART status** | Not on ART | 4 (5.8%) | 9 (10.6%) |
| | On ART | 40 (58.0%) | 33 (38.8%) |
| | Not applicable | 25 (36.2%) | 43 (50.6%) |
| **Organism** | *Escherichia coli* | 59 (85.5%) | 48 (56.5%) |
| | *Klebsiella* spp. | 4 (5.8%) | 20 (23.5%) |
| | Other Enterobacterales* | 6 (8.7%) | 17 (20.0%) |

Note

* Other organisms: *Proteus mirabilis* and *Enterobacter* spp. 3GC-S 3[rd] Generation Cephalosporin Susceptible, 3GC-R 3[rd] Generation Cephalosporin Resistant, ART Antiretroviral therapy, HIV Human Immunodeficiency Virus, CI Confidence Interval

5D VAS scores were available for 100/154 participants. Incomplete HRQoL data sets are due to patients dying before data could be collected.

Table 1 shows the characteristics of included participants. Median age of participants was 45.2 years, 55.8% of participants were HIV infected and 84.9% of HIV infected adults were on ART (81.3% in full cohort). The median duration of hospital stay was 9.0 days (Interquartile range [IQR]. 4.8–16.0) and the in-hospital case-fatality was 72/154 (46.8%). *E. coli* and *Klebsiella* spp. were the most frequently isolated organisms in the sub-study, as in the full cohort, causing 69.5% and 15.6% of BSI respectively. Just over half (55.2%) of organisms were 3GC-R.

The mean total direct medical cost across all participants was US$294.44 (SE: 23.6) (Table 2). The majority of these costs were accounted for by ward stay (US$157.60, SE:18.7) and investigations (US$99.41, SE: 5.7). The mean total direct medical cost of an admission with any 3GC-R organism was US$343.85 (SE: 33.7) and with any 3GC-S organism was US$233.59 (SE: 30.1). The mean total direct medical cost was US$110.27 (95%CR; 22.60–197.95) and higher amongst those who had a resistant BSI than those admitted with a sensitive BSI. For each resource-use category (ward stay; medications; investigations; procedures), the direct medical costs were higher for those who had 3GC-R organism than those who had 3GC-S organism (Table 2).

The mean total direct non-medical cost of an admission with any 3GC-R organism was US$129.81 (SE: 38.7) and with any 3GC-S organism was US$108.83 (SE: 23.8). In comparison to

**Table 2. Direct medical costs by 3<sup>rd</sup> generation cephalosporin susceptibility status (N = 154).**

| | N | Mean/SE (2019 US Dollars) | % of Total cost* | Mean Differences: 3GC-R v 3GC-S (95% CR)** |
|---|---|---|---|---|
| **3GC-S** | 69 | | | |
| Ward stay | | 121.30 (23.1) | 43.9 | |
| Medications | | 12.84 (4.1) | 6.5 | |
| Investigations | | 94.71 (8.9) | 47.6 | - |
| Procedures | | 4.72 (3.6) | 2.1 | |
| **Total** | | **233.59 (30.1)** | - | |
| **3GC-R** | 85 | | | |
| Ward stay | | 187.08 (26.7) | 46.9 | 65.78 (0.28, 131.28) |
| Medications | | 37.82 (10.0) | 8.8 | 24.98 (3.17, 46.79) |
| Investigations | | 103.23 (7.4) | 39.8 | 8.52 (-13.90, 30.93) |
| Procedures | | 15.72 (7.2) | 4.5 | 11.00 (-3.30, 25.29) |
| **Total** | | **343.85 (33.7)** | - | **110.27 (22.60, 197.95)** |

Note

*Mean of percentage at the participant level

**Bootstrapped estimates of mean differences and 95% credible interval (95% CR) 3GC-S 3<sup>rd</sup> Generation Cephalosporin Susceptible, 3GC-R 3<sup>rd</sup> Generation Cephalosporin Resistant, CR Credible Interval

those admitted with a sensitive BSI, those who had a resistant BSI incurred a mean additional direct non-medical cost of US$20.98 (95%CR; -36.47, 78.42). The mean total indirect cost of an admission with any 3GC-R organism was US$280.87 (SE: 129.1) and with any 3GC-S organism was US$125.39 (SE: 85.0). In comparison to those admitted with a sensitive BSI, those who had a resistant BSI incurred a mean additional indirect cost of US$155.48 (95%CR; -67.8, 378.78) (Table 3).

**Table 3. Direct non-medical and indirect costs by 3<sup>rd</sup> generation cephalosporin susceptibility status (N = 154).**

| | N | Dollars Mean/SE (2019 US) | Mean Differences: 3GC-R v 3GC-S (95% CR)* |
|---|---|---|---|
| **3GC-S** | 69 | | |
| Patient direct non-medical | | 45.91 (9.0) | |
| Patient indirect | | 55.08 (32.5) | |
| Family/carer direct non-medical | | 62.92 (20.0) | - |
| Family/carer indirect | | 70.31 (71.4) | |
| **Total direct non-medical** | | **108.83 (23.8)** | |
| **Total indirect** | | **125.39 (85.0)** | |
| **3GC-R** | 85 | | |
| Patient direct non-medical | | 55.77 (12.8) | 9.87 (-11.90, 31.63) |
| Patient indirect | | 83.58 (63.8) | 28.50 (-68.75, 125.75) |
| Family/carer direct non-medical | | 74.03 (31.4) | 11.11 (-36.22, 58.44) |
| Family/carer indirect | | 197.29 (113.2) | 126.98 (-96.04, 349.99) |
| **Total direct non-medical** | | **129.81 (38.7)** | **20.98 (-36.47, 78.42)** |
| **Total indirect** | | **280.87 (129.1)** | **155.48 (-67.80, 378.78)** |

Note

*Bootstrapped estimates of Mean differences and 95% credible interval (95% CR) 3GC-S 3<sup>rd</sup> Generation Cephalosporin Susceptible, 3GC-R 3<sup>rd</sup> Generation Cephalosporin Resistant, CR Credible Interval

**Table 4. Total societal costs, by 3rd generation cephalosporin susceptibility status (N = 154).**

|  | N | Mean/SE (2019 US Dollars) | Mean Differences: 3GC-R v 3GC-S (95% CR)* |
|---|---|---|---|
| **All** | 154 | 626.06 (93.1) | - |
| **3GC-S status** |  |  |  |
| Negative | 69 | 467.80 (83.9) | - |
| Positive | 85 | 754.53 (128.5) | 286.73 (43.76, 529.69) |

Note

*Bootstrapped estimates of Mean differences and 95% credible interval (95% CR)

3GC-S 3rd Generation Cephalosporin Susceptible, 3GC-R 3rd Generation Cephalosporin Resistant, CR Credible Interval

The mean total societal cost of a hospital admission across all participants was US$626.06 (SE 93.1) (Table 4). When stratified by 3GC susceptibility status however, the mean total societal cost of an admission with any 3GC-R organism was US$754.53 (SE: 128.5) and with any 3GC-S organism was US$467.80 (SE: 83.9). The mean total societal cost for those admitted with a resistant BSI was US$286.73 (95%CR; 43.76, 529.69) higher than those admitted with a sensitive BSI (Table 4).

Table 5 shows the findings of the multivariable analysis to investigate the independent effect of 3GC-R on total direct medical costs, indirect costs and societal costs. In model 1, after adjusting for age, sex and organism, 3GC-R was associated with increased direct medical and societal costs. In model 2, additionally adjusted for HIV status, 3CG-R was still associated with increased costs, but with credible intervals crossing zero.

Table 6 shows the EQ-5D utility and VAS scores for all participants, stratified by 3GC susceptibility status. Participants with 3GC-S infection had EQ-5D utility scores of 0.424 (SE: 0.06) and those with resistant infection had EQ-5D utility scores of 0.256 (SE: 0.08). Participants with resistant infections had a EQ-5D utility score that was therefore 0.167 (95% CR: -0.035, 0.300) lower than those with sensitive infections. EQ-5D utility scores generated using the UK tariff were lower, and the difference in EQ-5D utility scores between those with sensitive and resistant infections more pronounced. VAS scores were comparable in those with resistant and sensitive infection at 60.5 and 59.7 respectively, mean difference -0.8 (95% CR: -10.8, 9.1).

Table 7 shows the findings of the multivariable analysis to investigate the independent effect of 3GC-R on EQ-5D utility scores. In model 1, after adjusting for age, sex and organism, the EQ-5D utility score (generated using the Zimbabwean tariff) in those with a 3GC-R BSI was

**Table 5. Multivariable regression analysis of costs (N = 154)*.**

| 3GC status | Total direct medical cost (2019 US Dollars) | | Total indirect cost (2019 US Dollars) | | Total societal costs (2019 US Dollars) | |
|---|---|---|---|---|---|---|
|  | Model 1 | Model 2 | Model 1 | Model 2 | Model 1 | Model 2 |
|  | Coeff. (95% CR) | Coeff. (95% CR) | Coeff. (95% CR) | Coeff. (95% CR) | Coeff. (95% CR) | Coeff. (95% CR) |
| **3GC-S** | Ref. | Ref. | Ref. | Ref. | Ref. | Ref. |
| **3GC-R** | 88.45 (-3.37, 180.27) | 85.57 (-6.60, 177.75) | 105.23 (-129.00, 339.45) | 111.97 (-126.01, 350.95) | 229.57 (-17.96, 477.09) | 220.57 (-22.87, 464.01) |

*Note*: Model 1: additionally adjusted for age, sex and organism, Model 2: additionally adjusted for age, sex, organism and HIV status, CR = Credible interval, 3GC-S 3rd Generation Cephalosporin Susceptible, 3GC-R 3rd Generation Cephalosporin Resistant

Ref: reference category is organisms sensitive to third-generation cephalosporins

*Findings from Generalised Linear Model with Poisson distribution and Identity link function

**Table 6. Health-related quality of life outcomes by 3rd generation cephalosporin susceptibility status (N = 154).**

|  | N | Mean/SE | Mean Differences: 3GC-R v 3GC-S (95% CR) * |
|---|---|---|---|
| **3GC-S** |  |  |  |
| VAS score | 69 | 60.5 (5.2) | - |
| EQ-5D utility score (Zim Tariff) | 69 | 0.424 (0.06) |  |
| EQ-5D utility score (UK Tariff) | 69 | 0.320 (0.12) |  |
| **3GC-R** |  |  |  |
| VAS score | 85 | 59.7 (8.3) | -0.8 (-10.8, 9.1) |
| EQ-5D utility score (Zim Tariff) | 85 | 0.256 (0.08) | 0.167 (-0.035, 0.300) |
| EQ-5D utility score (UK Tariff) | 85 | 0.180 (0.18) | 0.140 (-0.091, 0.371) |

Note

*Bootstrapped estimates of Mean differences and 95% CR. CR = Credible interval, SE = Standard error, VAS = visual analogue scale, 3GC-S 3rd Generation Cephalosporin Susceptible, 3GC-R 3rd Generation Cephalosporin Resistant

0.159 (95% CR 0.034, 0.285) lower than those with a 3GC-S BSI. In model 2, additionally adjusted for HIV status, the EQ-5D utility score in those with a 3GC-R BSI was 0.149 (95%CR: 0.022, 0.276) lower than those with a 3GC-S BSI. The differences were comparable when the UK tariff was used to generate EQ-5D utility scores.

The annual direct medical and societal cost estimates for Malawi, associated with 3GC-R and 3GC-S *E. coli* BSI admissions are shown in S3 Table and in Fig 2. In 2016 it was estimated that the annual direct medical cost to provide hospital care for those admitted with *E. coli* BSI in Malawi was US$785,936 (95% CR, US$576,233, US$995,620) and of this, (75%) US$590,470 (95% CR, US$432,166, US$748,755), was from 3GC-R. By 2030, the annual direct medical cost is predicted to be US$1,216,243 (95% CR, US$891,726, US$1,540,732) (S3 Table).

The annual societal cost for hospitalisation with *E. coli* in 2016 was estimated to be US$1,657,354 (95% CR, US$915,861, US$2,398,828), of which US$1,271,954 (95% CR, US$678,553, US$1,865,336) (77%) was accounted for by 3GC-R. By 2030, the predicted societal cost for hospitalisation with *E. coli* BSI is US$2,564,771 (95% CR, US$1,417,303, US$3,712,209), with US$1,968,361 (95% CR, US$1,050,068, US$2,886,625), accounted for by 3GC-R infection.

The annual direct medical and societal cost estimates associated with 3GC-R and 3GC-S *Klebsiella* BSI admissions are shown in S4 Table and in Fig 3. For *Klebsiella* spp. it is estimated that in 2016, the annual direct medical cost was US$628,588 (95% CR, US$381,704, US

**Table 7. Multivariable regression analysis EQ-5D utility scores (N = 154)*.**

| Organism and 3GC susceptibility status | EQ-5D utility score–Zim Tariff | | EQ-5D utility score–UK Tariff | |
|---|---|---|---|---|
|  | Model 1 | Model 2 | Model 1 | Model 2 |
|  | Coeff. (95% CR) | Coeff. (95% CR) | Coeff. (95% CR) | Coeff. (95% CR) |
| **3GC-S** | Ref. | Ref. | Ref. | Ref. |
| **3GC-R** | -0.159 (-0.285, -0.034) | -0.149 (-0.276, -0.022) | -0.152 (-0.371, 0.067) | -0.135 (-0.358, 0.087) |

Note

Model 1: adjusted for age, sex and organism

Model 2: adjusted for age, sex, organism, and HIV status

Zim = Zimbabwe, CR = Credible interval, 3GC-S 3rd Generation Cephalosporin Susceptible, 3GC-R 3rd Generation Cephalosporin Resistant

Ref: reference category is organisms sensitive to third-generation cephalosporins

*Findings from Ordinary Least Squares estimator

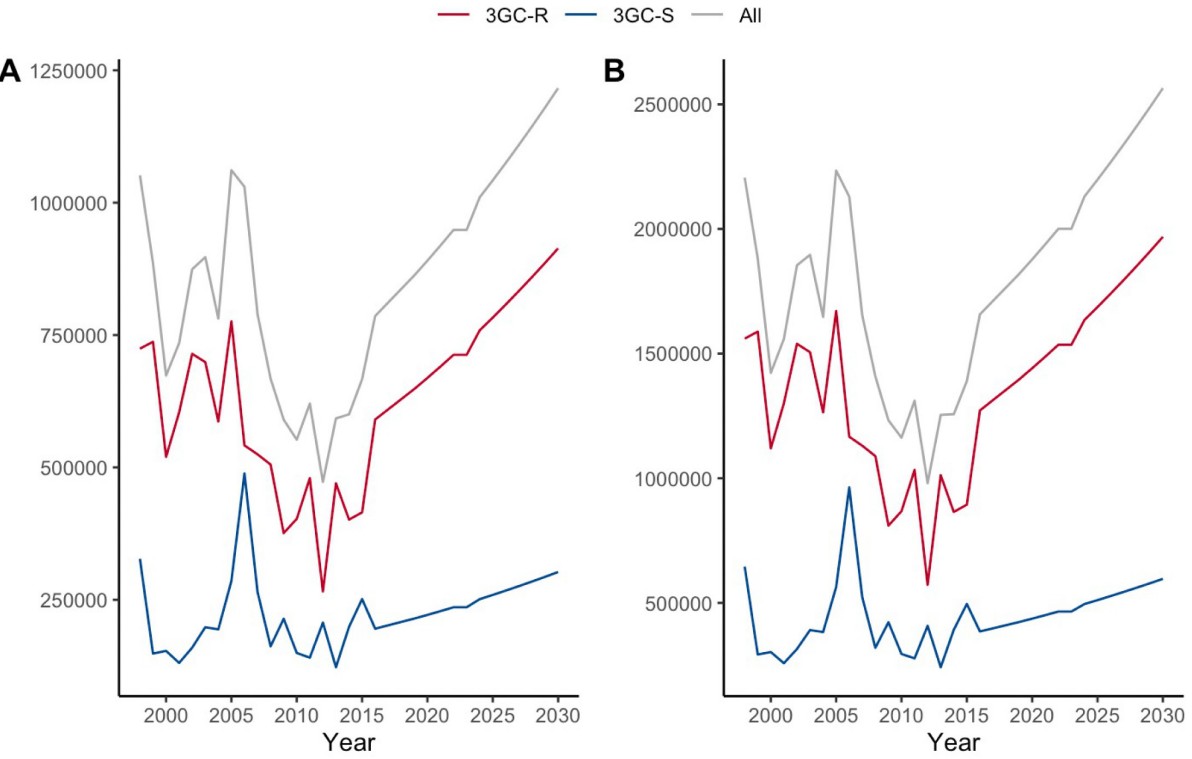

**Fig 2.** (A) Annual direct medical cost and (B) Annual societal cost estimates for *E. coli* BSI.

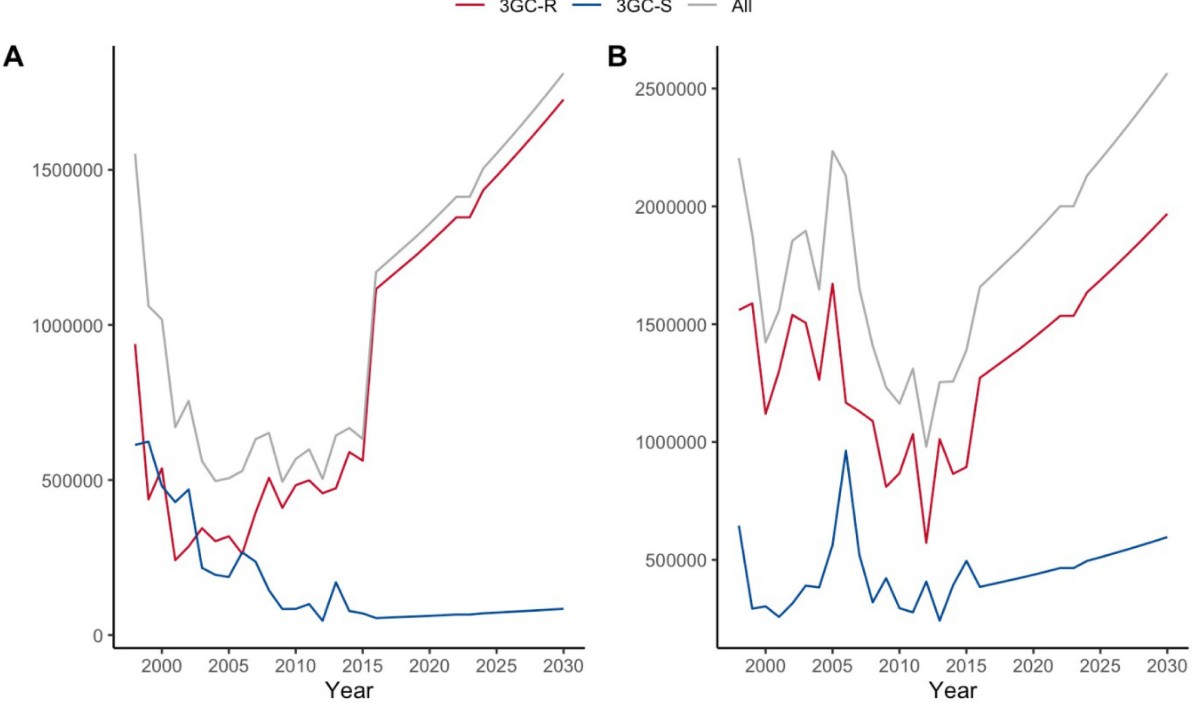

**Fig 3.** (A) Annual direct medical cost and (B) Annual societal cost estimates for *Klebsiella spp*. BSI.

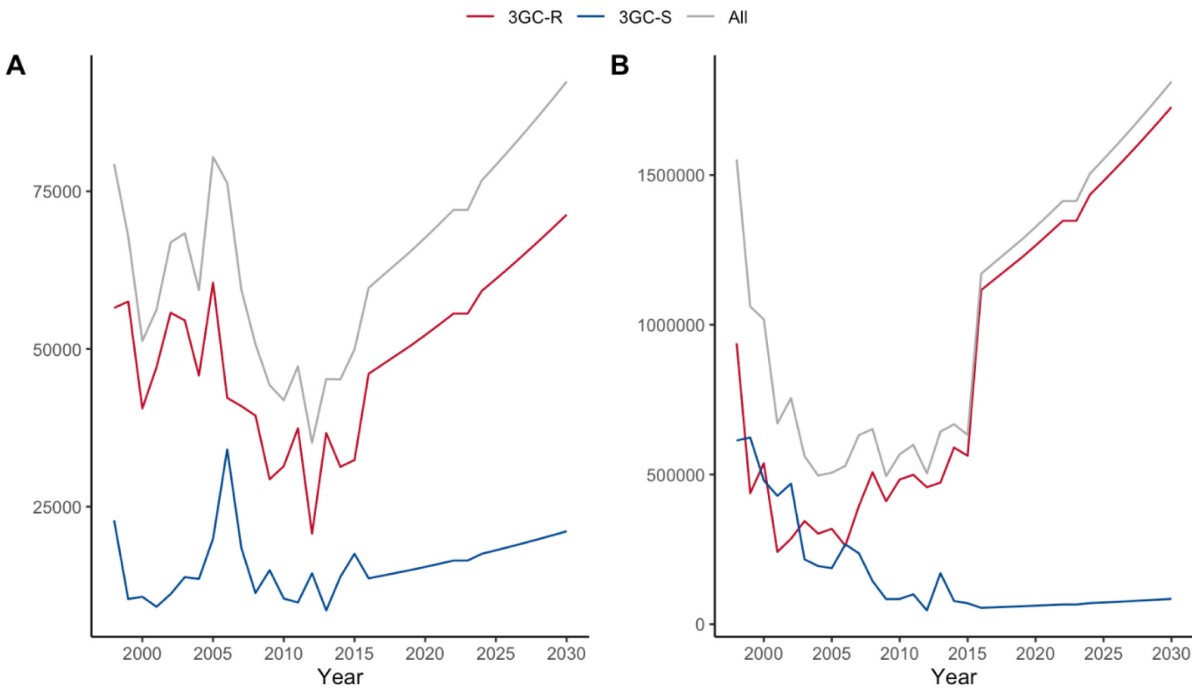

**Fig 4.** Annual QALYs lost for (A) *E. coli* and (B) *Klebsiella* BSI.

$875,488) and of this, 94% (US$589,993, 95% CR, US$367,813, US$830,188) was due to 3GC-R. By 2030, it is predicted that this will rise to an annual cost of US$972,746 (95% CR, US$590,691, US$1,354,827), with US$926,947 (95% CR, US$569,194, US$1,284,724) accounted for by 3GC-R.

The annual societal cost associated with hospitalisation with *Klebsiella* spp. BSI was estimated at US$1,170,714 (95% CR, US$502,571, US$1,838,874), of which 95% (US$1,116,030, 95% CR, US$488,074, US$1,744,002)) was associated with 3GC-R. By 2030, this annual societal cost is predicted to rise to US$1,811,691 (95% CR, US$777,734, US$2,845,675), with US$1,727,067 (95% CR, US$755,299, US$2,698,860) accounted for by 3GC-R.

The estimated total QALYs lost from *E. coli* and *Klebsiella* BSI are shown in S5 Table, stratified by 3GC-R status and in Fig 4. In 2016, it was estimated that *E. coli* infections accounted for 59,714 QALYs lost (95% CR, 59,696, 59,731), of which 46,078 (95% CR, 46,066, 46,091) (77%) were from 3GC-R. In the same time period, it is estimated that *Klebsiella* BSI accounted for 37,495 QALYs lost (95% CR, 37,478, 37,512), of which 33,897 (95% CR, 33,880, 33,913) (90%) were from 3GC-R. Projections for 2030 estimate that there will be 92,408 (95% CR, 92,381, 94,435) QALYs lost from *E. coli* BSI, of which 71,307 (95% CR, 71,288, 71,326) will be accounted for by 3GC-R. For Klebsilla BSI it is estimated that 58,024 (95% CR, 57,997, 58,050) QALYS will be lost, with 52,455 (95% CR, 52,430, 52,480) accounted for by 3GC-R.

## Discussion

Hospital admission with bloodstream infections caused by Enterobacterales, place a substantial financial burden on QECH, as well as on patients and their families. This burden was substantially higher amongst those who had infections that were resistant to third-generation cephalosporins, and patients admitted to hospital with resistant infections had poorer HRQoL. Using

the costs generated in this study, we estimated the economic burden of *E. coli* and *Klebsiella* BSI in Malawi, and found that 3GC-R accounts for more than 80% of the economic and health burden posed from these infections.

The average healthcare provider cost of managing patients with BSI in this cohort was US $294.44, comparable to estimates from the previous study of adult medical inpatients at QECH, in which the average cost of admission was US$313.65 (in 2014 prices) [7]. In this study, patients admitted with resistant infections were associated with an additional US $106.42 cost to the health provider than sensitive infections. To put this into context, the annual cost of providing anti-retroviral treatment in Malawi is approximately US$170 (in 2014 prices) [10]. AMR health provider cost data from sSA are extremely limited, with only one other study, from Senegal, reporting costs associated with 3GC-R infections. This study found the additional cost associated with 3GC-R was 100EUR (US$120.96 in 2012 prices) but did not report mean overall costs [21].

Ward stay and investigations accounted for the majority of the total direct medical costs of 3GC-R and 3GC-R infections. Only 7.8% of total costs were accounted for by spending on drugs, which is likely to reflect the WHO prequalification of medicines programme, which ensures availability of high quality medicines in Africa at reasonable prices. The costs of medications used in this study reflect the median costs paid by ministries of health in low income countries [22]. In the Senegalese study and in high income settings, the majority of the additional costs of 3GC-R infections come from the use of more expensive second and third-line antibiotics, such as carbapenems and aminoglycosides [21]. We found no difference in the proportion spent on medicines between resistant and sensitive infections, likely because these more expensive antibiotics were routinely available at QECH at the time of this study, however, as access to these drugs improves in Malawi, the cost of managing 3GC-R will rise even further.

Medical care at QECH is free of charge, but patients and their guardians inevitably incur some out-of-pocket costs as a result of a hospital admission, including for food and transportation. For patients and guardians together, the average spending on these non-medical items was US$120.41 and there was no substantial difference between 3GC-R and 3GC-S infection (mean difference US$20.98). However, this out-of-pocket cost is substantial considering that the majority of Malawians live on less than US$2 per day [23] and hospitalisation clearly poses a significant burden on household finances, potentially pushing those affected further into poverty. This is further compounded when the impact of these infection on household incomes is considered. The average indirect costs for families, incurred from loss of income was US$211.21 and this was significantly higher for 3GC-R than 3GC-S BSI, with a mean difference of US$155.48. The impact of these admissions on household finances is potentially devastating and savings for households from reducing 3GCR-BSI alone would be considerable.

Similarly, on a national scale we have estimated societal costs from 3GC-R of close to US$4 million annually by 2030. The impact of the costs of AMR are likely to have the greatest impacts on low-income countries such as Malawi, which have low overall healthcare expenditure, and mitigation strategies all the more important.

We carried out multivariable analysis to determine the independent effect of 3GC-R on costs. In models adjusted for age, sex and causative organism, 3GC-R BSI was associated with higher direct and societal costs as well as lower EQ-5D scores. HIV-status was then included as a model covariate, because previous work from QECH has shown that HIV-infected individuals have significantly higher health provider costs and significantly lower quality of life scores than HIV uninfected patients [7]. The estimated mean differences in HIV adjusted and HIV unadjusted models were very similar, although with wider credible intervals, suggesting that in

this cohort, HIV-infection has only a marginal effect on costs and that 3GC-R is likely to be of greater significance.

We used annual incidence data for the two most commonly isolated Gram negative organisms in this cohort, *E. coli* and *Klebsiella* spp [20], to estimate the economic burden of BSI in Malawi. The estimates generated predict an annual health provider spend of over $2million on just these two infections in 2030. Over 80% of this spend is accounted for by 3GC-R, therefore strategies which aim to reduce AMR are likely to have profound impacts on healthcare spending.

We demonstrate that patients with Enterobacterales bloodstream infection in Blantyre, Malawi, have poor quality of life, and contribute to the emerging evidence of high burden of disease from AMR in sub-Saharan Africa [24]. The most comprehensive global AMR burden estimates to date, suggest that for 3GC-R *E. coli*, 128 DALYs are lost per 100,000 population aged over 5 living in sSA [24]. For 3GC-R *Klebsiella pneumoniae*, this is 220 DALYs lost per 100,000. Our estimates are similar, with approximately 200 QALYs are lost per 100,000 Malawians, from each of 3GC-R *E. coli* and 3GC-R *Klebsiella* alone. European estimates suggested that AMR bacterial infections accounted for 170 DALYs per 100,000 population and that 3GC-R *E. coli* and *Klebsiella* accounted for over half of this burden [25]. This is more than the combined burden estimates for three major infectious diseases (influenza, TB and HIV) in Europe [26].

This study has a number of limitations. Primarily, the assumptions made in the calculation of economic burden will have generated conservative estimates for several reasons. Firstly, we have assumed that levels of 3GC-R remain at 2016 levels, when current trends suggest they will increase annually [20]. Secondly, in calculating the overall healthcare provider burden, we used costs estimated amongst adult admissions, and applied this to incidence data for patients of all ages. Paediatric admissions at QECH are likely to incur higher costs than adult admissions, because of the availability of more complex interventions (such as mechanical ventilation and central venous access). Primary costing studies on paediatric wards should therefore be a focus for future work, particularly since children carry a large proportion of the morbidity and mortality burden of 3GC-R. Our burden estimates have been derived from cost data collected from one hospital only. Expanding economic studies to other hospitals and healthcare settings within Malawi will help to generate improved estimates. Finally, although the EQ-5D tool is increasing used for health economic analysis in sub-Saharan African countries there is currently no Malawian tariff. Where no tariff exists for the country of interest it is widely accepted practice to use tariffs from another country, provided the two populations value health comparable [27]. We used the Zimbabwean tariff to derive EQ-5D utility scores and undertook a sensitivity analysis using the UK tariff.

Here, we demonstrate that 3GC-R BSI incur higher costs to the hospital and patients and lead to poorer HRQoL outcomes than 3GC-S infections. We have generated the some of the first AMR economic burden estimates for sub-Saharan Africa based on accurate, prospectively collected costing data. Strategies that reduce the incidence of 3GC-R infections could lead to significant cost savings to the hospital and patients, as well as improved QoL outcomes in those admitted to hospital.

## Supporting information

**S1 Text. Methods.**
(DOCX)

**S1 Table. Historical and projection population for Blantyre and Malawi.**
(DOCX)

**S2 Table. Observed and estimated sensitive and resistant infections.**
(DOCX)

**S3 Table. A.** Annual costs (2019 US Dollars) for *E. coli*–Mean costs. **B.** Annual costs (2019 US Dollars) for *E. coli*– 95% Upper Credible Interval. **C.** Annual costs (2019 US Dollars) for *E. coli*– 95% Lower Credible Interval.
(DOCX)

**S4 Table. A.** Annual costs (2019 US Dollars) for *Klebsiella* spp.–Mean costs. **B.** Annual costs (2019 US Dollars) for *Klebsiella* spp. - 95% Upper Credible Interval. **C.** Annual costs (2019 US Dollars) for *Klebsiella* spp. - 95% Lower Credible Interval.
(DOCX)

**S5 Table. A.** Annual QALYs lost due to *E. coli* and *Klebsiella* spp.—Mean QALYs. **B.** Annual QALYs lost due to *E. coli* and *Klebsiella* spp.– 95% Upper Credible Interval QALYs. **C.** Annual QALYs lost due to *E. coli* and *Klebsiella* spp.–Lower 95% Credible Interval QALYs.
(DOCX)

## Acknowledgments

The authors would like to thank the clinical staff at Queen Elizabeth Central Hospital

## Author Contributions

**Conceptualization:** Rebecca Lester, Christopher P. Jewell, Nicholas A. Feasey, Hendramoorthy Maheswaran.

**Data curation:** Rebecca Lester, James Mango, Jane Mallewa.

**Formal analysis:** Rebecca Lester, Hendramoorthy Maheswaran.

**Funding acquisition:** Rebecca Lester.

**Methodology:** Rebecca Lester, Christopher P. Jewell, Nicholas A. Feasey, Hendramoorthy Maheswaran.

**Supervision:** Christopher P. Jewell, David A. Lalloo, Nicholas A. Feasey, Hendramoorthy Maheswaran.

**Writing – original draft:** Rebecca Lester.

**Writing – review & editing:** James Mango, Jane Mallewa, Christopher P. Jewell, David A. Lalloo, Nicholas A. Feasey, Hendramoorthy Maheswaran.

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
