## [Decision Letter · Decision Letter 0]

21 Sep 2022

PGPH-D-22-01156

Individual and population level costs and health-related quality of life outcomes of third-generation cephalosporin resistant bloodstream infection in Malawi

Dear Dr. Lester,

Thank you for submitting your manuscript to PLOS Global Public Health. After careful consideration, we feel that it has merit but does not fully meet PLOS Global Public Health’s publication criteria as it currently stands. Therefore, we invite you to submit a revised version of the manuscript that addresses the points raised during the review process.

We look forward to receiving your revised manuscript.

Kind regards,

Malaisamy Muniyandi, Ph.D

Academic Editor

Journal Requirements:

1. We ask that a manuscript source file is provided at Revision. Please upload your manuscript file as a .doc, .docx, .rtf or .tex.

Additional Editor Comments (if provided):

Dear Authors

Greetings

Please go through the reviewers comments, the current version is not suitable for publication. This manuscript needs major revision, incorporate the reviewers comments and resubmit the revised version for further consideration.

Regards

Muniyandi

Reviewers' comments:

Reviewer's Responses to Questions

**Comments to the Author**

1. Does this manuscript meet PLOS Global Public Health’s publication criteria? Is the manuscript technically sound, and do the data support the conclusions? The manuscript must describe methodologically and ethically rigorous research with conclusions that are appropriately drawn based on the data presented.

Reviewer #1: Yes

Reviewer #2: Yes

2. Has the statistical analysis been performed appropriately and rigorously?

Reviewer #1: Yes

Reviewer #2: Yes

3. Have the authors made all data underlying the findings in their manuscript fully available (please refer to the Data Availability Statement at the start of the manuscript PDF file)?

Reviewer #1: No

Reviewer #2: Yes

4. Is the manuscript presented in an intelligible fashion and written in standard English?

Reviewer #1: Yes

Reviewer #2: Yes

5. Review Comments to the Author

Reviewer #1: The study explores and quantify the impact of third-generation cephalosporin resistant (3GC-R) bloodstream

infection (BSI) on economic and health related quality of life outcomes for adult patients in a teaching hospital located in Blantyre, Malawi. The article is well written and I would like to congratulate the authors for their hard work; however, there are some points that need clarification.

Major comments

❏ The title refers to Malawi, but I noticed you performed your research at the Queen Elizabeth Central Hospital (QECH) in Blantyre, so I don’t know if one hospital qualifies as a whole country (but I might be wrong). Likely, it is worth rewording the title and referring to the city and country (if the hospital is the largest in the town). Please also state whether it is a primary/secondary/tertiary care hospital or public/private throughout the introduction.

❏ I think readers are not entirely familiar with the specific studied city in Malawi. I recommend you provide some context about the city and how it is compared to the capital city in Malawi or the percentage of hospital patients attended compared to the whole country.

❏ How are the 3GC-R and 3GC-S infections defined based on MICs? Which thresholds did you use; CLSI, EUCAST?

❏ Why did you construct a parallel model including HIV infection? It is worth an in-depth explanation rather than just citing the paper. Likely, this will give us more context about the disease epidemiology of the targeted population. Are other specific features such as a high HIV prevalence embedded in the population?

❏ I have seen that you modelled and projected the estimates using the chosen hospital as the reference. First, you should state that you projected the costs using the reference hospital in the abstract.

❏ I noticed you described 154 patients on line 205 (results), but it is not equivalent to what is illustrated in Figure 1. Would you mind adding the exact sample size throughout the methods section? Does that differ by the outcome? Is there any missing data, how Is it handled and was it at random? Was the study fully randomised?

❏ If missing data were due to death, I presume the costs are highly underestimated. How would you handle this?

❏ Table 1; would you mind inserting all the abbreviations fully described in Table 1’s notes? This might include ART, HIV, 3GC-S and R, CI, and IQR. Also, you have enough space to write Escherichia coli instead of E. coli. Could you please state the total number of observations next to Table 1’s title (e.g., {N=154})?

❏ Tables 2-7; state the total number of obs in the title. Refer to the acronyms/abbreviations in full in notes (this might include 3GC, SE, CR, etc.). Coef should be written as “Coeff.”. The “Ref” term should be written with a dot by the end and describe it fully in notes.

❏ Table 5. Could you pls replicate the analyses for total indirect costs? Would you mind presenting the p-values? You might want to add rows to refer to the other variables included in the analysis (you can put a tick box to show what is included or not, and present the full results in the supplementary material).

❏ What were multivariable analyses employed? Linear methods using linear regressions? Did you test the distributions? Skewness, multicollinearity of independent variables, homoscedasticity? Did you use robust SE? How the values for the time series were predicted (till 2030)?

❏ I wonder if using the Zimbabwean tariff for the EQ-5D impacts the estimates, why you chose Zimbabwe and not any other, and how that might be linked to Malawi. What is the rationale for using UK and Zimbabwe’s tariffs? Likely, it is worth having a word in the limitation’s subsection.

❏ Did you inflate and transform to 2019 USD the costs you mentioned in the discussion from other articles?

❏ The abstract should be rewritten, accounting for some of the comments I have made throughout. For instance, define the multivariable methods used; you mentioned the mean health provider costs in results but did not define it in methods, etc. Also, state the currency throughout the abstract (2019 USD).

Sensitivity analyses.

❏ I would expect to see variations in cost estimates, given how the data were reported (indexes used) or the targeted population. For instance, you did not include children or ppl aged <18 years. How might the estimates vary if we can account for that? You used the Zimbabwean and UK tariffs for EQ-5D; would it change if a Malawian index were constructed? I know that it's not available everywhere, but how might it affect the estimates?

Minor comments

❏ Spell out the LMICs abbreviation once first mentioned.

❏ Spell out TB once first mentioned.

❏ I think the sentence highlighting the software used should be moved to the last part of the statistical analyses subsection.

❏ Line 83.. it says “to informed” should it be to inform?

❏ HIV and ART should be spelt out once first mentioned

❏ IQR, spell this out.

❏ Table 2; the “n” should be written in capital letters to be consistent with the rest.

❏ I will use a subtitle in the methods section for the societal costs. Societal costs= direct and indirect medical and non-medical costs; so then you describe them all. I did not realise they were calculated as the sum till I got to the statistical analyses’ subsection.

❏ Why do you use 2019 USD and international dollars?

❏ Could you pls expand the labels and legends in Figures 2-4 and make the lines thicker? Figure 4 should be similar to the previous ones (legends on top favour the reading, no need to repeat the legends in both panels). Also, A and B are bolded in Figures2-3 and not in Figure 4. It should be great to make them consistent.

❏ Could you please present the 95% CI for the predicted values in Figures 2-4.

Reviewer #2: Even the manuscript is technically sound and it represents an important additional information on cost impact of AMR in a Low Income Country, some major revision should been made:

1. There is not any discussion on EQ-5D limitations use in the african context. It must be included in the methodology section and in the discussion one.

2. The societal cost in terms of out of pocket costs and loss of income seem to be hiden in the findings. Those are not discussed at all. It would be relevant to be included some comments around them, given the Country context.

3. It is lost the reflexion on the rational use of 3GC in order to improve the health outcomes given the findings (are there some clinical guidelines? is there any surveillance of AMR system?).

6. PLOS authors have the option to publish the peer review history of their article (what does this mean?). If published, this will include your full peer review and any attached files.

**Do you want your identity to be public for this peer review?** For information about this choice, including consent withdrawal, please see our Privacy Policy.

Reviewer #1: No

Reviewer #2: No

---

## [Editor Report · Decision Letter 1]

9 Mar 2023

PGPH-D-22-01156R1

Individual and population level costs and health-related quality of life outcomes of third-generation cephalosporin resistant bloodstream infection in Blantyre, Malawi

Dear Dr. Lester,

Thank you for submitting your manuscript to PLOS Global Public Health. After careful consideration, we feel that it has merit but does not fully meet PLOS Global Public Health’s publication criteria as it currently stands. Therefore, we invite you to submit a revised version of the manuscript that addresses the points raised during the review process.

Please check the Pdf version of Table 7: Multivariable regression analysis EQ-5D utility scores (N=154)*

We look forward to receiving your revised manuscript.

Kind regards,

Malaisamy Muniyandi, Ph.D

Academic Editor
---

## [Editor Report · Decision Letter 2]

6 Apr 2023

Individual and population level costs and health-related quality of life outcomes of third-generation cephalosporin resistant bloodstream infection in Blantyre, Malawi

PGPH-D-22-01156R2

Dear Dr Lester,

We are pleased to inform you that your manuscript 'Individual and population level costs and health-related quality of life outcomes of third-generation cephalosporin resistant bloodstream infection in Blantyre, Malawi' has been provisionally accepted for publication in PLOS Global Public Health.

Best regards,

Malaisamy Muniyandi, Ph.D

Academic Editor